# Rapid and Nondestructive Evaluation of Wheat Chlorophyll under Drought Stress Using Hyperspectral Imaging

**DOI:** 10.3390/ijms24065825

**Published:** 2023-03-18

**Authors:** Yucun Yang, Rui Nan, Tongxi Mi, Yingxin Song, Fanghui Shi, Xinran Liu, Yunqi Wang, Fengli Sun, Yajun Xi, Chao Zhang

**Affiliations:** 1State Key Laboratory of Crop Stress Biology for Arid Areas, College of Agronomy, Northwest A&F University, Xianyang 712100, China; 2Key Laboratory of Wheat Biology and Genetic Improvement on Northwestern China, Ministry of Agriculture, Xianyang 712100, China

**Keywords:** wheat leaves chlorophyll, drought stress, machine learning, regression model, high-resolution spectral imaging, high-throughput phenotypic identification

## Abstract

Chlorophyll drives plant photosynthesis. Under stress conditions, leaf chlorophyll content changes dramatically, which could provide insight into plant photosynthesis and drought resistance. Compared to traditional methods of evaluating chlorophyll content, hyperspectral imaging is more efficient and accurate and benefits from being a nondestructive technique. However, the relationships between chlorophyll content and hyperspectral characteristics of wheat leaves with wide genetic diversity and different treatments have rarely been reported. In this study, using 335 wheat varieties, we analyzed the hyperspectral characteristics of flag leaves and the relationships thereof with SPAD values at the grain-filling stage under control and drought stress. The hyperspectral information of wheat flag leaves significantly differed between control and drought stress conditions in the 550–700 nm region. Hyperspectral reflectance at 549 nm (r = −0.64) and the first derivative at 735 nm (r = 0.68) exhibited the strongest correlations with SPAD values. Hyperspectral reflectance at 536, 596, and 674 nm, and the first derivatives bands at 756 and 778 nm, were useful for estimating SPAD values. The combination of spectrum and image characteristics (L*, a*, and b*) can improve the estimation accuracy of SPAD values (optimal performance of RFR, relative error, 7.35%; root mean square error, 4.439; R^2^, 0.61). The models established in this study are efficient for evaluating chlorophyll content and provide insight into photosynthesis and drought resistance. This study can provide a reference for high-throughput phenotypic analysis and genetic breeding of wheat and other crops.

## 1. Introduction

Wheat (*Triticum aestivum* L.) is a staple food crop crucial for global food security. However, wheat crops experience many abiotic stresses, including low temperatures, drought, high temperatures, and dry and hot winds, which can strongly affect growth, development, and productivity [1,2]. Drought is one of the most severe abiotic stresses worldwide and can significantly reduce the number of tillers, grains per spike, and 1000-grain weight of wheat [3,4]. In 2021, both the United States and Brazil suffered historic severe droughts, which increased global food prices to the highest cost in a recent decade. Therefore, effective monitoring of the impact of drought stress during wheat growth is essential for improving yields, varieties, and food security.

Drought stress decreases crop chlorophyll content, destroys photosynthetic machinery, inhibits growth, and ultimately reduces yield [5,6]. In addition, flag leaf chlorophyll content can reflect the growth status of plants [7,8]. Chlorophyll is involved in capturing light energy and converting it into chemical energy during photosynthesis. It is fundamental for plant photosynthesis and directly determines capacity, net primary productivity, and carbon budget (Appendix A) [9,10,11]. The synthesis of chlorophyll in plants is affected by a variety of factors, including temperature, water, disease, and other stresses. Water shortage of plant leaves will affect the synthesis of chlorophyll, accelerate the decomposition of chlorophyll, reduce the absorption capacity of light energy, and inhibit photosynthesis [12,13]. Meanwhile, plants also regulate carotenoid synthesis in response to stress and play a role in light energy capture and light protection during photosynthesis [14,15]. Crop chlorophyll content may be increased or decreased under drought stress [16,17,18], and the degree of change is closely related to drought resistance [19,20,21,22]. Therefore, monitoring flag leaf chlorophyll content can provide key information regarding wheat photosynthesis and drought resistance. Traditional methods for determining chlorophyll content include spectrophotometry and the use of hand-held chlorophyll content meters [23]. However, such methods have several disadvantages, such as leaf destruction and low-throughput, which are not conducive to the measurement of wheat chlorophyll content on a large scale. Compared to traditional methods, hyperspectral imaging (HSI) can determine plant chlorophyll content rapidly, nondestructively, and efficiently [24,25]. In addition, hyperspectral images contain rich spectral information that can be used for precision agriculture research and the establishment of complex mathematical models [26,27]. In recent years, the application of (HSI) technology to plant monitoring has developed rapidly. Such methods are based on the principle that different substances exhibit different absorption and reflection characteristics in distinct spectral bands [28]. The optical characteristics of the leaves include light reflected from the leaf’s external surface and internal leaf structures, light absorbed by internal leaf chemicals (e.g., various leaf pigments), and light transmission (Appendix A) [29]. The visible band (400–750 nm) reflects the synthesis and decomposition of chemical pigments critical for plant photosynthetic activity [30]; the near-infrared band (750–1000 nm) characteristics depend on leaf internal structures, absorption by water, and scattering processes [31]. Stress changes the pigments, water, and structures of plant leaves, and shows different spectral characteristics [32]. Certain hyperspectral curve characteristics, such as chlorophyll absorption troughs at 450 and 650 nm, a reflection peak near 540 nm, and typical red-edge effects in the range of 700–760 nm, reflect plant chlorophyll content and growth status [33,34,35,36]. Furthermore, drought stress causes leaf spectral reflectance to increase in the visible region, while chlorophyll content tends to decrease [37,38]. These aspects of HSI make it a feasible approach for estimating plant chlorophyll content. Extensive research has focused on developing models based on the spectral index for estimating chlorophyll content [24,39,40,41,42,43]. However, several problems have been identified in terms of general applicability and stability, which limits the application of the estimation models to other systems [44,45,46]. Compared to the spectral index, full-band hyperspectral information represents various aspects of physiology more comprehensively; combined with machine learning methods, such information can substantially improve model accuracy [47,48,49]. Several studies have monitored the physiological characteristics of plants based on information from sensitive bands [50,51]. However, the small number of sensitive bands does not adequately represent all hyperspectral information. Moreover, most studies used few wheat varieties and ignored the heterogeneous inter-multiple variety. Therefore, the applicability of previous models to other systems is limited [44], and the models would be ineffective for large-scale chlorophyll content and drought resistance assessments.

In the present study, we collected 335 wheat varieties (total 2010 leaf samples) under drought stress and control conditions at the filling stage to explore the relationship between hyperspectral image characteristics and chlorophyll content. A machine learning method was used to construct models to establish a rapid, nondestructive, accurate, and widely applicable method for assessing wheat chlorophyll content, photosynthesis, and drought resistance.

## 2. Results

### 2.1. Chlorophyll Changes in Wheat Leaves under Different Soil Moisture Conditions

SPAD values reflect changes in crop chlorophyll content and can thus be used to closely track photosynthesis. In this study, the median and upper quartile (Q3) SPAD values indicated that the chlorophyll content of several varieties tended to increase under drought stress, while lower quartile (Q1), lower whisker, and small outlier values suggested that the chlorophyll content of other varieties significantly declined after drought stress (Figure 1). These results indicate that the chlorophyll content of wheat changed significantly during drought stress and that the extent of the change varied among wheat varieties. Thus, we can infer that different wheat varieties exhibit differences in photosynthetic capacity under drought stress, which is closely related to drought resistance.

### 2.2. Hyperspectral Characteristics of Wheat Leaves under Different Soil Moisture Conditions

#### 2.2.1. Spectral Reflectance Characteristics under Different Soil Moisture Conditions

The leaf is the most important organ for photosynthesis in wheat, and the spectral reflection curves of leaves reflect their growth status. After drought stress, the spectral reflectance of several wheat varieties increased significantly in the visible region (400–700 nm) (Figure 2A). The mean spectral reflectance of wheat leaves markedly increased between 600–700 nm, which represents an absorption trough for chlorophyll (Figure 2C).

#### 2.2.2. Spectral First Derivative Values under Different Soil Moisture Conditions

The first derivative can reduce the influence of noise in hyperspectral information. In the present study, spectral first derivative values of wheat leaves significantly differed between control and drought stress conditions, with increases and decreases seen in the regions between 550–650 and 700–750 nm, respectively, under drought stress (Figure 2B,D). Thus, we can infer that differences in the first derivative values were related to differences in the chlorophyll content and growth status of wheat leaves.

#### 2.2.3. Spectral Images under Different Soil Moisture Conditions

Single-band hyperspectral images are often used to observe sensitive areas of the plant spectral response. Single-band hyperspectral images corresponding to the characteristics of the leaf hyperspectral curve were collected (Figure 2E). The hyperspectral images obtained under drought stress conditions at 450 and 500 nm (in the blue band), and at 970 nm (water absorption zone), did not significantly differ from the images of the controls. In contrast, significant differences were observed for the green and red bands (550, 600, 650, and 700 nm). These results indicate that the green and red bands represent sensitive areas for monitoring wheat chlorophyll content and growth status under drought stress.

### 2.3. Correlation Analysis between Hyperspectral Characteristics and SPAD Values

Correlation analysis revealed strong negative correlations between spectral reflectance and SPAD values in the visible region (400–735 nm), particularly at 549 and 708 nm (correlation coefficients of −0.64 and −0.61, respectively). However, no significant correlations were detected in the near-infrared region (750–1000 nm) (Figure 3A). The correlation coefficient between the first derivative of the spectrum and SPAD values was −0.61 at 541 nm, 0.68 at 735 nm, and −0.44 near 946 nm. Meanwhile, the first derivative and SPAD values were positively correlated near 600 nm under control conditions but were negatively correlated under drought stress (Figure 3B). These results indicate that the chlorophyll content was strongly correlated with hyperspectral features in the visible region but only weakly correlated in the near-infrared region. In addition, the spectral first derivative had a stronger correlation with SPAD values compared to reflectance.

### 2.4. The Characteristic Bands Identified with the Successive Projections Algorithm for Estimating SPAD Values

The successive projections algorithm (SPA) can eliminate redundant hyperspectral information and improve modeling speed and efficiency, and it is widely used for extracting characteristic bands. The hyperspectral reflectance bands at 536, 596, and 674 nm (Figure 4A), and the first derivative bands at 756 and 778 nm (Figure 4B), were the optimal bands for estimating SPAD values. These results indicate that hyperspectral information at the green, red, and red-edge bands is closely related to wheat SPAD values, which can therefore be used to monitor wheat leaf chlorophyll content.

### 2.5. Principal Component Analysis of Hyperspectral Information

Principal component analysis (PCA) is a dimensionality reduction technique widely used for hyperspectral data analysis that can reduce dimensionality with minimal information loss. Principal component loading represents the correlation coefficient between a principal component and the original variable. In the present study, PCA was used to reduce the dimensionality of the hyperspectral reflectance data and first derivatives, and the loadings of the first three principal components were calculated. Figure 4C,D present the 3D spatial distribution of leaf spectral reflectance and the first derivative, respectively. The first three principal components were dispersed under drought stress but more aggregated under control conditions (Figure 4C,D). The variance explained by the first principal component of spectral reflectance (68.92%) was higher than that explained by the first derivative (33.05%). These results indicate that drought stress significantly affected leaf blade hyperspectral reflectance, and the collinearity of the hyperspectral first derivative was lower than that of reflectance. Together, the first two principal components of spectral reflectance explained more than 94% of the variance in the data; principal component 1 had high loadings at 446 and 695 nm, while principal component 2 had high loadings at 624, 695, and 770 nm (Figure 4E). The first three principal components of the first derivative of the spectrum explained nearly 70% of the variance; principal component 1 had high loadings at 494, 684, and 722 nm; principal component 2 had high loadings at 523, 616, and 703 nm; and principal component 3 had a high loading at 852 nm (Figure 4F). These results indicate that the drought stress spectra were strongly correlated with the principal components at these wavebands.

### 2.6. Estimation of SPAD Values Based on Regression Analysis

#### 2.6.1. Estimation of SPAD Values Based on Spectral Characteristics

Henceforth, CA-R and CA-FD are used to denote the highest correlations of spectral reflectance and the first derivative, with SPAD values, while (SPA-R) and (SPA-FD) are used to indicate characteristic band reflectance and the first derivative, as extracted by SPA. Full-R and Full-FD are used to denote full-band reflectance and the first derivative, respectively, whereas PCA-R and PCA-FD are used to indicate spectral reflectance and the first derivative principal components, respectively, derived from the PCA.

The regression model results indicated that the RFR model using Full-R as the independent variable had the best optimal fit (R^2^ = 0.60, RMSE = 4.495, RE = 7.35%) among all models. The SLR model based on the CA-R had the worst fit (training set, R^2^ = 0.401, RMSE = 5.549, RE = 9.17%) due to the underfitting of the data (Table 1). The models based on Full-R were better than those based on Full-FD (Table 1, Figure 5E,F); although, the models built with CA-FD and SPA-FD provided better fits than CA-R and SPA-R (Figure 5A–D). The models built with SPA-R, SPA-FD, PCA-R, and PCA-FD were superior to the SLR models but inferior to the Full-R and Full-FD models (Table 1). The LASSO and RR models based on SPA-R, SPA-FD, PCA-R, and PCA-FD exhibited underfitting (Table 1).

The fit of the SLR model established with the first derivative at 735 nm (CA-FD) was better than that of the model based on spectral reflectance at 549 nm (CA-R), the data in the former case exhibited a stronger linear relationship with the SPAD values (Figure 3C–F). The dotted lines in Figure 3E,F, Figure 6, and Figure 7 are the 1:1 fit line between the predicted and measured SPAD values, which indicate that the RFR model based on Full-R provided the best fit (Figure 6C). The deviation between the actual and 1:1 fit lines of the models based on Full-R (Figure 6A–C) was smaller compared with the models based on Full-FD (Figure 6D–F). These results indicate that the first derivative transformation can enhance the signal-to-noise ratio of the characteristic bands and improve model accuracy, which may result in the loss of some full-band hyperspectral information. At the same time, the feature band spectrum did not adequately represent whole-leaf hyperspectral information, while the full-band hyperspectral image appears to have more potential for chlorophyll content evaluation.

The LASSO, RR, and RFR models built with the Full-R demonstrate outstanding stability and accuracy in predicting the SPAD values under different soil moisture conditions (RFR models R^2^ > 0.5; Figure 7). The LASSO model prediction performance of SPAD values under drought stress is the best with R^2^ 0.569, RMSE 5.159, and RE 9.28%. As a result, the hyperspectral reflectance in predicting the SPAD values of the wheat leaves under drought stress can provide a robust result and has the potential to be used in drought resistance identification in the future.

To further verify the reliability of models, the SLR models constructed by 549 nm reflectance and 735 nm first derivative were used for SPAD visual mapping (Figure 8). These results are consistent with leaf RGB images (Figure 2E), which indicates that the HSI monitoring SPAD values and drought stress are reliable.

#### 2.6.2. Estimation of SPAD Values Based on Spectral and Image Characteristics

CIEL*a*b* is a visually consistent color model in which the color difference is closer to the actual perceived color difference and is the best way to express the range of perceived colors by the human eye. The L*, a*, and b* values have significant correlations with SPAD values (r = −0.591, −0.164, −0.600; Table 2). Furthermore, we combined L*a*b* features and spectral characteristics to construct RFR models, which indicates that the Full-R combined with L*, a*, and b* has the optimal SPAD values prediction effect (R^2^ = 0.61, RMSE = 4.439, RE = 7.35%). Overall, the combination of spectral characteristics and L*a*b* features improve the estimation accuracy for SPAD values (Table 3).

## 3. Discussion

### 3.1. Feasibility of Estimating Chlorophyll Content of Wheat Leaves Using Hyperspectral Information

Previous studies have demonstrated that plant leaf hyperspectral curves exhibit significant differences in the visible region under stress conditions. For example, Gang et al. (2010) reported a significant correlation between chlorophyll content and spectral characteristics in the visible region (400–680 nm) in castor bean seedlings under salt stress [52]. In the present study, the hyperspectral reflectance curves of wheat leaves under different soil moisture conditions varied remarkably in the visible region (500–700 nm; Figure 2A,C) and were strongly correlated with SPAD values (Figure 3A). These findings indicate that the hyperspectral reflectance curves of wheat leaves under stress conditions were closely related to the chlorophyll content in the visible region (400–700 nm), and, therefore, adequately represent leaf chlorophyll content.

Studies of tea leaves have shown that the hyperspectral reflectance between 552–555 nm and 707–735 nm is strongly correlated with the chlorophyll content [53]. Under drought stress, the spectral first derivative of apple leaves in the visible region (513–539, 564–585, 694, and 699 nm) was also closely related to the chlorophyll content [54]. In the present study, leaf spectral reflectance between 512–603 and 700–723 nm, and the spectral first derivative between 485–542 and 719–760 nm, were strongly correlated with SPAD values (Figure 3A,B). These results highlight the feasibility of using hyperspectral characteristics to estimate chlorophyll content accurately and efficiently under drought stress.

### 3.2. Models for Estimating SPAD Values in Wheat Leaves

Our SPAD values estimation model based on full-band hyperspectral information exhibited good performance. In a previous study using characteristic bands and full-band hyperspectral reflectance to estimate wheat yield under low-temperature stress, a full-band principal component regression (PCR) model (R^2^ = 0.854, root mean square error of prediction 625.7) had the best stability [55]. In the present study, the LASSO, RR, and RFR models based on Full-R were superior to all other models and had R^2^ and RMSE values of 0.585 and 4.578, 0.585 and 4.575, and 0.60 and 4.495, respectively (Figure 6A–C, Table 1). These findings demonstrate that models based on Full-R had better stability and accuracy.

The first derivative can reduce the influence of noise and improve the signal-to-noise ratio of the data. Previous chlorophyll content evaluations of pitaya stems revealed that first derivative transformation can enhance the sensitivity of feature bands to chlorophyll content, and a model based on the first derivative of characteristic bands was optimal (R^2^ = 0.625, RMSE = 0.048) [56]. A study estimating maize yield found that full-band RR (R^2^ = 0.54, RMSE = 2.58) and support vector regression (SVR; R^2^ = 0.53, RMSE = 2.69) models performed better than a full-band first derivative model (R^2^ = 0.41, RMSE = 3.51 and R^2^ = 0.49, RMSE = 2.95, respectively); however, RFR models showed the opposite result [57].

The present study demonstrated that the combining of spectral characteristics with image features (L* a* b*) can improve the estimation accuracy of SPAD values (Table 1 and Table 3). These results showed that the fit for diverse data sets varied among the models. Thus, it is necessary to determine which models are most suitable for particular data sets to achieve the best prediction performance. In future studies, we can utilize multiplicative scatter correction (MSC), standard normal variate transformation (SNV), and other algorithms to preprocess hyperspectral data and apply deep learning to enhance stability and accuracy.

### 3.3. Utility of Hyperspectral Reflectance for Monitoring Wheat Growth and Evaluating Drought Resistance under Drought Stress

The hyperspectral characteristics of wheat leaves were closely related to wheat growth status under different soil moisture conditions. The hyperspectral information of soybean leaves can achieve the accurate estimation for chlorophyll content (R^2^ = 0.94, RMSE = 0.201) [49]. Similarly, previous research showed that 617, 675, and 818 nm are the optimal bands for estimating the chlorophyll content of diseased peach fruit [58]. Using the continuous wavelet transform (CWT) to estimate wheat SPAD values under low-temperature stress, a previous study found that 553, 727, 728, 729, and 734 nm are SPAD-sensitive bands, and that spectral reflectance at 553 nm can accurately estimate SPAD values (R^2^ = 0.7444, RMSE = 7.359) [59]. One study using multiple feature selection methods to determine SPAD values in pepper leaves reported that the characteristic bands were concentrated within the regions of 415.4–431.5, 526.7–676.2, and 839.3–979.2 nm [60]. Furthermore, 548, 718, and 727 nm were the best wave bands for estimating chlorophyll content in grafted cucumber seedling leaves [61]. Taken together, these studies clearly indicate that hyperspectral information is closely related to chlorophyll content; however, it was still necessary to establish a stable, reliable, and universally applicable model for evaluating diverse species, and for within-species analyses, under different growth conditions. In the present study, full-band hyperspectral reflectance of combined L*, a*, and b* accurately estimated SPAD values (R^2^ = 0.61, RMSE = 4.439, RE = 7.35%; Table 3). The spectral reflectance bands at 536, 549, 596, 674, and 708 nm, and the first derivative bands at 735, 756, and 778 nm, were SPAD characteristic bands (Figure 3A,B and Figure 4A,B). Nonetheless, many bands within the 446–770 nm region were closely related to drought stress (Figure 4E,F). Using SLR based on 549 nm reflectance and 735 nm first derivative to assess SPAD values in the leaf level found that the HSI can monitor leaf chlorophyll content and drought stress reliably (Figure 8). In conclusion, plant HSI has great potential for evaluating leaf chlorophyll content under stress conditions; however, studies including large-scale varieties experiencing different stress conditions are necessary to establish stable and reliable models.

In most cases, drought stress reduces leaf chlorophyll content, although several studies have shown that some plants exhibit increased chlorophyll content under drought stress [62,63]. We found that the leaf SPAD values of several wheat varieties increased under drought stress (Appendix A), which may be related to the variation in drought resistance seen among different wheat varieties. The characteristics of the hyperspectral curves of different wheat varieties under drought stress also reflected differences in drought resistance among varieties. Therefore, future evaluation models of wheat drought resistance could be applied to aid genetic breeding of crop resistance.

## 4. Materials and Methods

### 4.1. Plant Material and Growth Conditions

The study area was located in the National Dryland Plant Variety Rights Trading Center in Yang Ling District (108°4′E, 34°16′N), China, in which new crop varieties are tested. In total, 335 wheat varieties (Appendix A) were planted in a steel frame shed on 21 October 2021. The shed had good ventilation and fine sandy loam soils, to which we applied a compound fertilizer (N-P_2_O5-K_2_O) at 750 kg/hm^2^ prior to sowing. The soil drilling method was used to measure soil water content at a depth of 0.5 m. Beginning at the jointing stage of wheat growth, the relative soil water contents of the control and drought stress treatments were maintained at 75 ± 5% and 50 ± 5%, respectively. The calculation method of relative soil water content is consistent with previous research [64], and uses the formula below:$$\mathrm{Soil}\;\mathrm{water}\;\mathrm{content}\;(\%)=\frac{\mathrm{weight}\;\mathrm{of}\;\mathrm{moist}\;\mathrm{soil}\;-\;\mathrm{weight}\;\mathrm{of}\;\mathrm{dried}\;\mathrm{soil}}{\mathrm{weight}\;\mathrm{of}\;\mathrm{dried}\;\mathrm{soil}}\;\times \;100\;\mathrm{Soil}\;\mathrm{water}\;\mathrm{holding}\;\mathrm{capacity}\;(\%)=\frac{\mathrm{weight}\;\mathrm{of}\;\mathrm{water}\;\mathrm{in}\;\mathrm{saturated}\;\mathrm{soil}}{\mathrm{weight}\;\mathrm{of}\;\mathrm{dried}\;\mathrm{soil}}\;\times \;100\;\mathrm{Relative}\;\mathrm{soil}\;\mathrm{water}\;\mathrm{content}\;(\%)=\frac{\mathrm{soil}\;\mathrm{water}\;\mathrm{content}}{\mathrm{soil}\;\mathrm{water}\;\mathrm{holding}\;\mathrm{capacity}}\;\times \;100$$

The filling stage is the most rapid stage of wheat grain development, and photosynthetic capacity during this period strongly affects yield. Meanwhile, drought stress at the wheat filling stage has a significant impact on photosynthesis and wheat yield [65,66]. Therefore, it is important to estimate the chlorophyll content of wheat under drought stress in the filling stage for screening the drought-resistant wheat varieties. The experimental data were collected after flowering for 14 days during April and May 2022.

### 4.2. Hyperspectral Image Acquisition

Leaf hyperspectral images were acquired using the Pika L system (Resonon Inc., Bozeman, MT, USA), which provides 281 spectral bands (2.1-nm spectral resolution) in the visible near-infrared region (400–1000 nm) and has 900 spatial channels. We collected six flag leaves samples from each variety, which were kept fresh [67] and transported to the laboratory for hyperspectral image collection under controlled lighting conditions. Before image acquisition, the dark current was removed and whiteboard calibration was conducted. During the collection stage, the leaf blades were wiped clean and kept flat. The average of three biological replicates was used as the hyperspectral data for subsequent analysis.

### 4.3. SPAD Values Measurement

A SPAD-502 Plus chlorophyll meter (Minolta, Tokyo, Japan) was used to obtain soil plant analysis development (SPAD; i.e., chlorophyll) values for wheat flag leaves, and this meter has been shown to correlate well with actual chlorophyll levels and to be a reliable method for nondestructive chlorophyll detection [68,69,70]. SPAD values of the tip, middle, and base of the leaf were obtained (five replicates each), and the average values were used in the analysis.

### 4.4. Hyperspectral Image Preprocessing

Hyperspectral images were obtained at the same time as SPAD values acquisition, which was preprocessed using the Spectronon Pro software (version 2.116)bundled with the Resonon Pika L acquisition system. A Savitzky–Golay filter was used to smooth the hyperspectral images. The first derivative was calculated using Spectronon Pro. In this study, Spectronon Pro was used to acquire CIEL*a*b* color space images and calculate L*, a*, and b* values.

### 4.5. Data Processing

Prism 9 (GraphPad Software, San Diego, CA, USA) and Python 3.6 (Python Software Foundation, Beaverton, OR, USA) were used for correlation and regression analyses. The training and testing sets included 80% and 20% of the data (*n* = 670), respectively. Values were calculated for model evaluation, coefficient of determination (R^2^), root mean square error (RMSE), and relative error (RE). Four regression models were tested: simple linear regression (SLR) [71], least absolute shrinkage and selection operator (LASSO) regression [72], ridge regression (RR) [73], and random forest regression (RFR) models [74].

## 5. Conclusions

In this study, the HSI of large-scale wheat varieties under different soil moisture conditions was used to determine the accuracy of rapid leaf chlorophyll content estimation models. The most sensitive bands with respect to chlorophyll content estimation were in the visible band (400–780 nm), and correlation analysis revealed that the optimal bands were located near 541, 549, 708, and 735 nm. The SPA indicated that the reflectance bands at 536, 596, and 674 nm were the optimal bands for estimating SPAD values. The first derivative bands at 756 and 778 nm were the most useful for estimating the relative chlorophyll content. Combining spectral characteristics and L*a*b* features can improve the accuracy of estimating the SPAD values of drought-stressed wheat (RFR model optimal performance: R^2^ = 0.61, RMSE = 4.439, RE = 7.35%). The technical method established in this study has great potential for evaluating chlorophyll content and stress resistance.

## Figures and Tables

**Figure 1 ijms-24-05825-f001:**
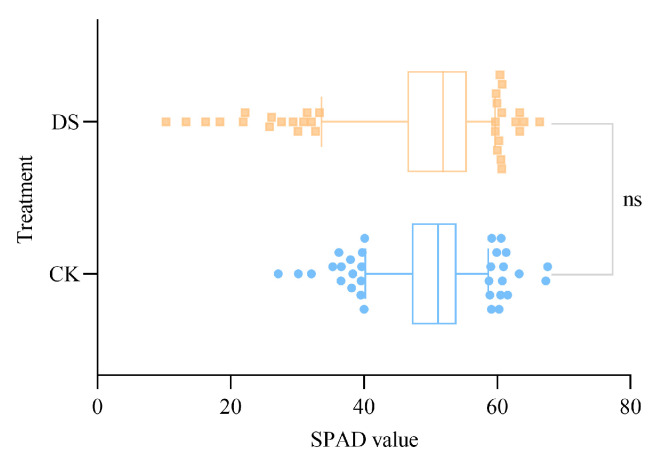
SPAD values of 335 varieties of wheat leaves under different soil moisture conditions. CK, control; DS, drought stress; whisker, 5–95%; ns, no statistically significant difference in SPAD values between the CK and DS treatments (*p* > 0.05).

**Figure 2 ijms-24-05825-f002:**
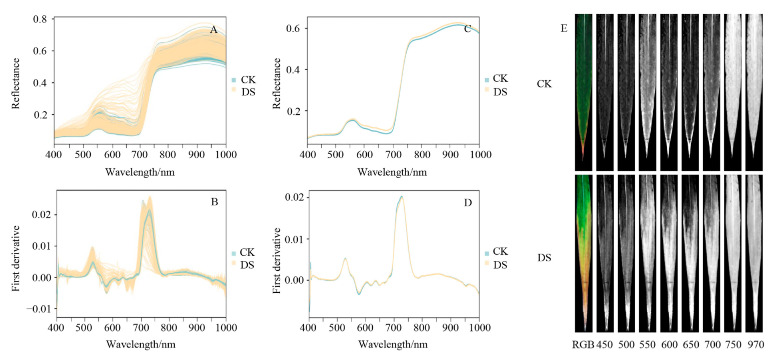
Hyperspectral curves and single-band hyperspectral images of wheat leaves under different soil moisture conditions. Hyperspectral reflectance and first derivative curves of leaves from 335 wheat varieties under control (CK) and drought stress (DS) conditions; (**A**,**B**) hyperspectral reflectance curves of leaves and first derivative values; (**C**,**D**) mean hyperspectral reflectance curves of leaves and first derivative values; (**E**) single-band hyperspectral images; hyperspectral band region (400–1000 nm); hyperspectral images obtained using the Pika L hyperspectral imaging system; RGB images (red = 640 nm, green = 550 nm, blue = 460 nm).

**Figure 3 ijms-24-05825-f003:**
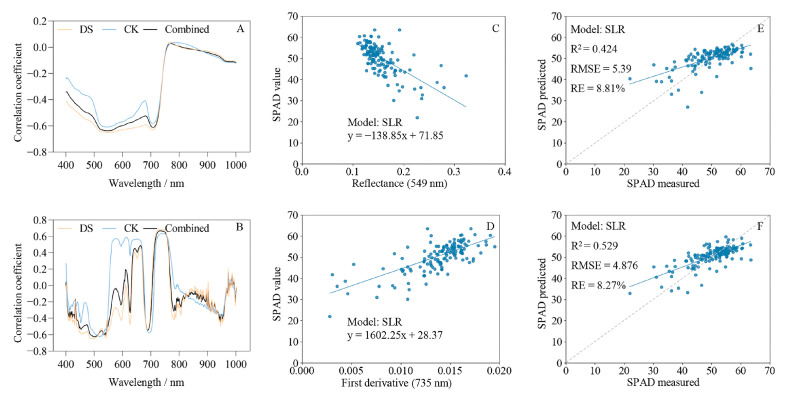
Correlation analysis and fitting results between leaf hyperspectral and SPAD values. (**A**,**B**) Correlations of spectral reflectance and the first derivative with SPAD values; (**C**,**D**) simple linear regression (SLR) analysis based on spectral reflectance at 549 nm and the spectral first derivative at 735 nm; (**E**,**F**) fitting results of predicted and measured values of SPAD based on reflectance at 549 nm and the first derivative at 735 nm. The gray dotted line is the 1:1 fit line between the predicted and measured values. Pearson correlation analysis of hyperspectral reflectance and first derivatives with SPAD values in CK, DS, and combined data sets was performed using Prism 9. Python 3.6 was used for SLR analysis of the data. CK, control; DS, drought stress; combined, all data.

**Figure 4 ijms-24-05825-f004:**
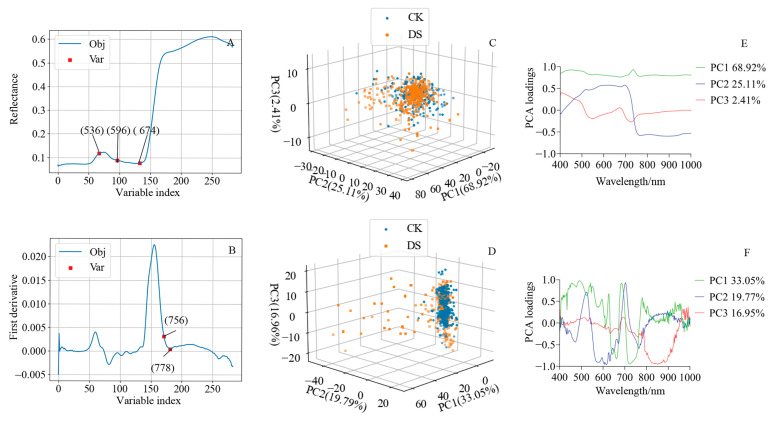
SPA feature band selection and PCA dimension reduction. (**A**,**B**) Spectral reflectance and first derivative bands extracted by SPA; (**C**,**D**) 3D spatial distribution of the first three principal components of spectral reflectance and the first derivative; (**E**,**F**) the first three principal components’ loadings of spectral reflectance and the first derivative. Python 3.6 was used to extract SPAD characteristic bands. Prism 9 was used to calculate PCA loadings. The percentage is the proportion of the variance explained by each principal component. SPA, successive projections algorithm; PCA, principal component analysis. Obj, first calibration object; Var, selected variables. CK, control; DS, drought stress.

**Figure 5 ijms-24-05825-f005:**
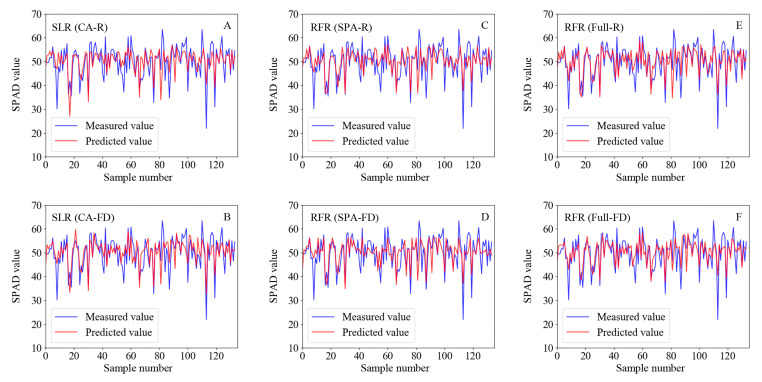
Comparison of predicted and measured values SPAD based on different data sets and models. (**A**,**B**) Simple linear regression (SLR) model based on CA-R and CA-FD; (**C**,**D**) random forest regression (RFR) model based on SPA-R and SPA-FD; (**E**,**F**) RFR model based on Full-R and Full-FD. CA-R/FD, reflectance/first derivative with the highest correlation with SPAD values; SPA-R/FD, reflectance/first derivative of a characteristic band extracted through SPA; Full-R/FD, full-band reflectance/first derivative. CK, control; DS, drought stress.

**Figure 6 ijms-24-05825-f006:**
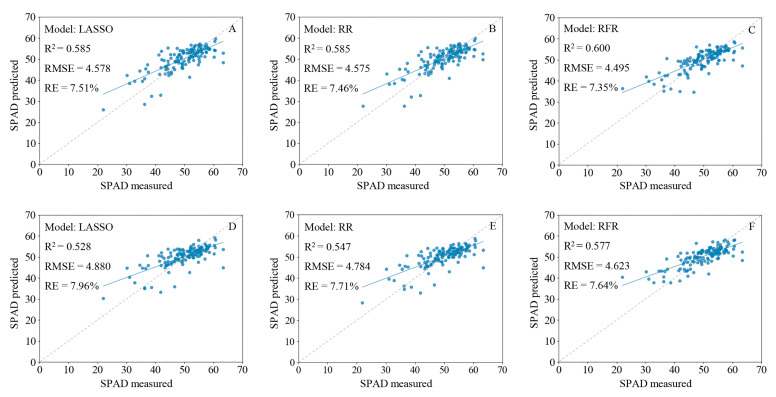
Fitting results of predicted and measured SPAD values of wheat leaves based on a full-band hyperspectral reflectance and first derivative model. (**A**–**C**) Least absolute shrinkage and selection operator (LASSO) regression, ridge regression (RR), and random forest regression (RFR) models built with full-band hyperspectral reflectance; (**D**–**F**) LASSO regression, RR, and RFR models built with full-band hyperspectral first derivative. The gray dotted line is the 1:1 fit between the predicted and measured SPAD values, and the blue solid line is the actual fit between predicted and measured values.

**Figure 7 ijms-24-05825-f007:**
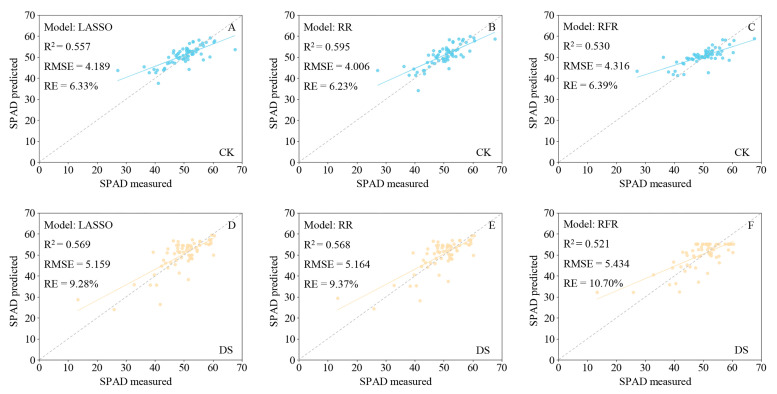
Fitting results of predicted and measured SPAD values of wheat leaves based on a full-band hyperspectral reflectance model under different soil moisture conditions. (**A**–**C**) Least absolute shrinkage and selection operator (LASSO) regression, ridge regression (RR), and random forest regression (RFR) models under the control condition; (**D**–**F**) LASSO regression, RR, and RFR models under the drought stress condition. The blue and yellow solid lines are the actual fit between predicted and measured values, and the gray dotted line is the 1:1 fit between the predicted and measured SPAD values. CK, control; DS, drought stress.

**Figure 8 ijms-24-05825-f008:**
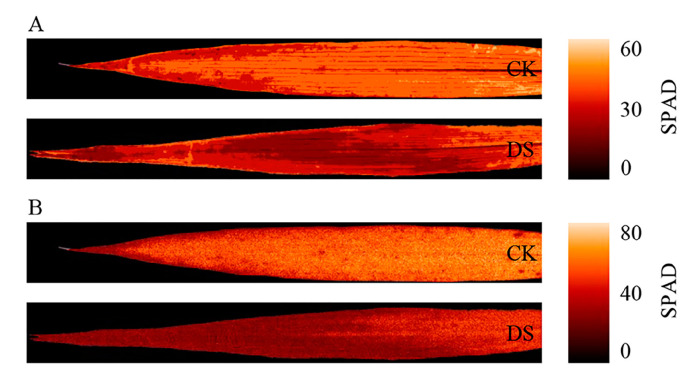
SPAD values map at the leaf level estimated by the characteristic reflectance and first derivative. (**A**,**B**) SPAD values map at the leaf level estimated by the 549 nm reflectance and 735 nm first derivative. CK, control; DS, drought stress. ENVI 5.3 (ITT, Visual Information Solutions, Boulder, CO, USA) was used to obtain the SPAD values map.

**Table 1 ijms-24-05825-t001:** The different models fitting results between SPAD and various spectral characteristics.

Data Set	Model	Training Set		(n = 536)	Testing Set		(n = 134)
		R^2^	RMSE	RE	R^2^	RMSE	RE
CA-R (549 nm)	SLR	0.401	5.549	9.17%	0.424	5.390	8.81%
CA-FD (735 nm)	SLR	0.426	5.434	9.01%	0.529	4.876	8.27%
SPA-R	LASSO	0.405	5.532	9.12%	0.417	5.430	8.86%
	RR	0.405	5.532	9.12%	0.417	5.426	8.85%
	RFR	0.478	5.183	8.60%	0.478	5.134	8.40%
SPA-FD	LASSO	0.407	5.523	9.25%	0.518	4.934	8.49%
	RR	0.407	5.523	9.26%	0.517	4.939	8.51%
	RFR	0.510	5.023	8.57%	0.510	4.974	8.47%
PCA-R	LASSO	0.484	5.150	8.39%	0.571	4.655	7.64%
	RR	0.488	5.130	8.20%	0.580	4.607	7.47%
	RFR	0.555	4.788	7.87%	0.555	4.739	7.89%
PCA-FD	LASSO	0.454	5.298	8.62%	0.496	5.045	7.90%
	RR	0.454	5.298	8.64%	0.497	5.039	7.99%
	RFR	0.560	4.755	7.74%	0.560	4.714	7.62%
Full-R	LASSO	0.587	4.609	7.43%	0.585	4.578	7.51%
	RR	0.586	4.617	7.45%	0.585	4.575	7.46%
	RFR	0.600	4.535	7.40%	0.600	4.495	7.35%
Full-FD	LASSO	0.528	4.929	8.12%	0.528	4.880	7.96%
	RR	0.548	4.824	7.81%	0.547	4.784	7.71%
	RFR	0.579	4.653	7.63%	0.577	4.623	7.64%

*n*, number of samples; R^2^, coefficient of determination; RMSE, root mean square error; RE, relative error (percentage); SLR, simple linear regression; LASSO, least absolute shrinkage and selection operator; RR, ridge regression; RFR, random forest regression. CA-R/FD, reflectance/first derivative with the highest correlation with SPAD values; SPA-R/FD, reflectance/first derivative of a characteristic band extracted through SPA; PCA-R/FD, principal components of reflectance/first derivative; Full-R/FD, full-band reflectance/first derivative; SPA, successive projections algorithm; PCA, principal component analysis.

**Table 2 ijms-24-05825-t002:** Relationships among SPAD, L*, a*, and b* of wheat leaves.

Variable	SPAD	L*	a*	b*
SPAD	1			
L*	−0.591 **			
a*	−0.164 **	0.438 **		
b*	−0.600 **	0.912 **	0.378 **	1

** Significant at the 0.01 probability level.

**Table 3 ijms-24-05825-t003:** Results of the RFR models based on spectrum and L*a*b* characteristics data set.

Data Set	Training Set (*n* = 536)			Testing Set (*n* = 134)		
	R^2^	RMSE	RE	R^2^	RMSE	RE
(CA-R) + L*a*b*	0.486	5.14	8.52%	0.486	5.095	8.23%
(CA-FD) + L*a*b*	0.519	4.973	8.19%	0.519	4.925	8.18%
(SPA-R) + L*a*b*	0.502	5.061	8.38%	0.501	5.021	8.27%
(SPA-FD) + L*a*b*	0.51	5.019	8.37%	0.51	4.972	8.13%
(PCA-R) + L*a*b*	0.46	5.271	8.76%	0.46	5.218	8.29%
(PCA-FD) + L*a*b*	0.584	4.626	7.47%	0.584	4.579	7.48%
(Full-R) + L*a*b*	0.61	4.478	7.30%	0.61	4.439	7.35%
(Full-FD) + L*a*b*	0.578	4.661	7.64%	0.578	4.617	7.58%
L*a*b*	0.435	5.39	9.06%	0.434	5.346	8.38%

L*a*b* represent L*, a*, and b* color component values.

## Data Availability

All data are contained within the article.

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
