# Peer review of "Rapid and Nondestructive Evaluation of Wheat Chlorophyll under Drought Stress Using Hyperspectral Imaging"

_ijms, 2023, doi:10.3390/ijms24065825_

Round 1

Reviewer 1 Report

I kindly invite the authors to check the points mentioned below:

-Please replace the keywords with more suitable keywords because the keywords are repeating with title words. 

Hyperspectral imaging (HSI) should be abbreviated in the first place where hyperspectral imaging was taken, and then it should be repeated in the text.

Is there a reason for the stress applied during grain-filling?

Correction needed on line 69: [21, 31] [12, 32-34].

Correction needed on line 75: [40] [41]

Line 92: (p > 0.05), P should be change to italic.

Lie 378: drought stress treatments were maintained at 75 ± 5% and 50 ± 5%, respectively, % of whatØŸ From the field capacity? available Water content? should be specified.

The quality of pictures and figures should be improved.

In Table 1, remove ** at the top of 1.

Would it not be better to compare the Hyperspectral imaging with the amount of chlorophyll obtained by solvent extraction?

Style of writing the references in the text and references section is not uniform.

Moreover, I kindly invite the authors to check punctuation and spelling because -there are some errors due to a lack of consistency in the manuscript.

-General revisions of English are suggested.

My recommendation is "Major Revision.

Reviewer 2 Report

Well done. Kindly, find the comments in the attached file. 
